# Effects of attachment designs on clear aligner tooth movement: A finite element analysis

Khaled Alsharif[1], Peter Ngan[1], Guoqiang Guan[1], Egon Mamboleo[2], Abdelhak Ouldyerou[2], Ali Merdji[3], Osama M. Mukdadi[2]*

1 Department of Orthodontics, School of Dentistry, West Virginia University, Morgantown, West Virginia, United States of America, 2 Department of Mechanical, Materials and Aerospace Engineering, West Virginia University, Morgantown, West Virginia, United States of America, 3 Department of Mechanical Engineering, Faculty of Science and Technology, University of Mascara, Mascara, Algeria

* sam.mukdadi@mail.wvu.edu

## Abstract

This study investigates the impact of different attachment shapes and configurations on the displacement, stress, and strain profiles of maxillary first molar during clear aligner-based orthodontic treatment. A subject-specific 3D maxillary model was developed from CBCT imaging, incorporating cortical and trabecular bone, periodontal ligament (PDL), teeth, attachments, and aligner geometry. Five attachment shapes square, rectangle, trapezoid, ellipse, and semicircle were analyzed in single and dual (buccal-lingual) configurations across four clinically relevant movements: mesialization, intrusion, extrusion, and rotation. Finite-element simulation results indicated that flat-shaped attachments (rectangular and trapezoidal) generated the greatest crown displacement but induced higher PDL strain (up to 0.390 mm/mm) and localized bone stress (7.11 MPa), particularly at the root apex and alveolar crest. Curved attachments provided more diffused load distribution but significantly reduced movement efficiency. Dual attachments improved root engagement and bodily displacement in all movement types, mitigating undesired tipping and enhancing force symmetry, albeit with elevated strain. Rotational control was most influenced by attachment geometry, with flat designs producing greater angular movement. Overall, attachment shape and placement exert a substantial influence on orthodontic biomechanics during aligner therapy. The findings underscore the need for evidence-based attachment protocols tailored to specific movement goals and patient risk profiles. These insights can guide clinicians toward optimizing clear aligner treatments for improved movement precision, minimized biological risk, and enhanced treatment outcomes in complex orthodontic cases.

## Introduction

Clear aligner therapy has become one of the most widely used alternatives to traditional fixed orthodontic appliances [1]. This rise in popularity is due in large part to its

**Data availability statement:** All relevant data are within the paper.

**Funding:** The author(s) received no specific funding for this work.

**Competing interests:** The authors have declared that no competing interests exist.

ability to provide patients with a more aesthetic, comfortable, and hygienic option for tooth movement. Clear aligners are particularly effective in treating mild to moderate malocclusions, and their increased adoption has led to continual advancements in materials, manufacturing methods, and treatment protocols [2]. However, as the demand for treating more complex cases with aligners grows, challenges associated with difficult tooth movements such as translation of individual teeth have become more apparent. Translation of posterior teeth such as molar mesialization and distalization, is essential in en masse movement of all posterior teeth or space closure following premolar extractions. This movement is biomechanically challenging due to the large root surface area of molars and their role in maintaining anchorage, often resulting in tipping, rotation, or undesired displacement of adjacent teeth.

To enhance control during complex tooth movements, composite attachments are commonly bonded to the surface of teeth [3]. These attachments increase aligner retention, improve force application, and provide mechanical leverage to aid in specific movements. However, despite their widespread clinical use, the biomechanical effectiveness of different attachment shapes and placement locations remains inadequately studied. Most clinical applications of attachments are based on empirical knowledge and manufacturer guidelines rather than rigorous quantitative evidence [4]. The decision to use a particular attachment shape or placement strategy often lacks biomechanical validation, especially regarding how these factors affect stress distribution, tooth displacement direction, and strain accumulation within the PDL and supporting alveolar bone.

Finite element analysis (FEA) is a computational technique that enables the simulation of complex mechanical systems by dividing them into smaller, manageable finite elements. In orthodontics, FEA provides a powerful framework to investigate internal stress and strain fields that cannot be observed clinically. Through simulation, FEA allows researchers to assess the effectiveness of various treatment parameters such as aligner material, thickness, attachment geometry, and boundary conditions [5,6]. FEA has been shown to be particularly useful in modeling the biomechanics of clear aligners and their interaction with tooth and bone structures, offering insights that can inform clinical treatment planning [7,8].

The present study aims to provide a biomechanical basis for attachment selection during molar mesialization in clear aligner therapy. By offering a detailed analysis of how shape and placement affect stress, strain, and displacement profiles, this study can assist clinicians in optimizing attachment protocols to improve movement efficiency while reducing the risk of adverse effects such as anchorage loss, root resorption, and excessive tipping. The findings support a more evidence-based approach to clear aligner treatment planning, particularly in cases that require precise and controlled tooth movement [9].

## Materials and methods

### Three-dimensional model construction

A subject-specific three-dimensional model of the maxilla was developed from cone-beam computed tomography (CBCT) scans acquired from a 20-year-old female

patient at the West Virginia University Department of Orthodontics. A research protocol (WVU IRB Protocol #2108398881) received an approval from the Institutional Review Board (IRB) at West Virginia University for the period (September 22, 2021 – September 21, 2026). Scanning was performed using the Carestream CS9300 system (Rochester, NY), generating 440 axial slices with a spatial resolution of 563×563 pixels and a slice thickness of 0.3 mm. The DICOM files were first accessed on October 10, 2021 and imported into 3D Slicer (Version 5.03) for segmentation [10].

Segmentation thresholds were defined using Hounsfield unit (HU) ranges to isolate anatomical features. The cortical and trabecular bone of the maxilla was segmented using a HU range of 262.05–1448.75, while the teeth were segmented using 965.71–3257.39 HU. Manual refinement using region-growing, thresholding, and paint tools was required to correct voxel fusion between neighboring teeth and bone. The segmented structures were exported as STL files for geometric optimization [11].

In Meshmixer (Autodesk, Version 3.8.0, San Francisco, CA) [12], noise artifacts were removed, and smoothing operations were performed. The trabecular bone was modeled by offsetting the internal surface of the maxilla inward by 2 mm resulting in a concentric shell structure that reflects actual bone layering. The PDL was reconstructed by isolating the tooth roots and extruding them outward by 0.2 mm, consistent with anatomical thicknesses reported in literature [13].

The design of the clear aligner and attachments was performed using ANSYS SpaceClaim (2024 R1, ANSYS, Canonsburg, PA). Five distinct attachment shapes were modeled: square (2 mm × 2mm), rectangle (2 mm × 4 mm), trapezoid (2 mm width with 14° side bevels), quad ellipse, and semicircle (1 mm radius). These attachments were placed on the buccal surface of the first molar in all simulations as seen in Fig 1, with some models including lingual attachments to evaluate dual-surface influence [9]. The clear aligner geometry was generated by combining the teeth and attachments into a composite model and offsetting the outer surface by 0.8 mm to replicate clinical aligner thickness.

## Mesh generation and preprocessing

The full assembly including bone, teeth, PDLs, attachments, and clear aligner was imported into ANSYS Mechanical (2024 R1) [14] for finite element analysis (FEA). Each element was discretized using 8-node tetrahedral elements (SOLID185) to provide accurate modeling of curved anatomical structures. Mesh smoothing and auto-surface wrapping were completed in SpaceClaim to ensure proper manifold geometry [15].

A mesh convergence study was conducted to optimize the trade-off between computational accuracy and processing time. The PDL was meshed with 0.4 mm element size to capture fine structural detail, while teeth and attachments were meshed at 0.6 mm. The cortical and trabecular bone were assigned to a mesh size of 1.0 mm. The clear aligner was meshed with an element size of 0.5 mm. The accuracy of the FE model was determined by evaluating the error in calculating maximum stress and maximum nodal displacement within the model. A 5% error threshold was set to evaluate FE simulation convergence and accuracy. Convergence was confirmed when subsequent refinements led to a convergence error of 4.87% in estimating maximum von Mises stresses, and a convergence error of 0.84% in calculating maximum nodal displacements [16].

## Boundary conditions and contact definitions

The maxilla was constrained in all six degrees of freedom at the midpalatal suture, replicating rigid skeletal anchorage. The defined displacements for the molar were applied to simulate aligner-driven loading in each test scenario. Simulations were performed under quasi-static assumptions using the Newton-Raphson solver, assuming full equilibration of force before proceeding to the next simulated aligner stage [17].

Contact conditions were implemented to represent physical interactions within the system. A bonded condition was assigned between the root surfaces and the PDL and between cortical and trabecular bone to simulate rigid physiological attachment. Frictional contact was defined between the aligner and the teeth, as well as between the aligner and

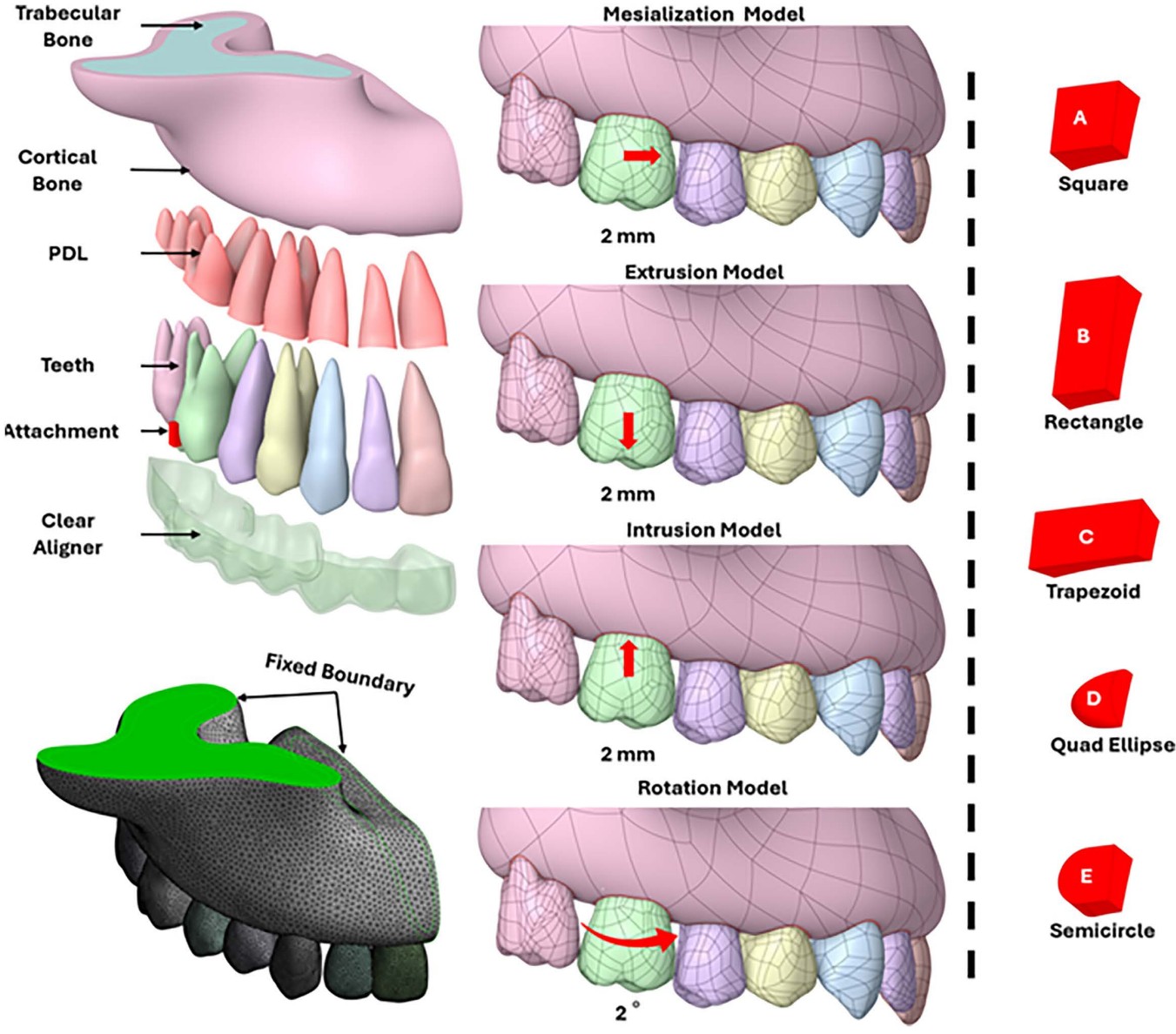

**Fig 1. Simulation models and attachment designs.** (Top Left) 3D model of the cortical and trabecular bone, PDL, teeth, attachment, and clear aligner; (Bottom Left) Fixed boundary conditions; (Center) Simulation movement patterns; (Right) Attachment shapes **A)** Square, **B)** Rectangle, **C)** Trapezoid, **D)** Quad Ellipse, and **E)** Semicircle.

attachments, with a coefficient of friction set to 0.2. Load ramping was applied to frictional interfaces to aid convergence and replicate gradual aligner force application [17–19].

Four clinically relevant orthodontic movements were simulated individually: mesialization, intrusion, extrusion, and rotation of the maxillary first molar. For mesialization, extrusion, and intrusion movements, a prescribed displacement of 0.2 mm was applied to the crown center of the molar in the respective direction. For rotational movement, a 2° angular displacement was applied about the vertical axis of the tooth. These displacements represent typical aligner-induced steps and allowed for comparative analysis of attachment efficacy [20].

## Material properties

All materials were modeled as isotropic, homogeneous, and linearly elastic based on data from prior validated FEA studies in orthodontics. The cortical bone was assigned to a Young's modulus of 13,700 MPa and a Poisson's ratio of 0.30. Trabecular bone was assigned a modulus of 1,370 MPa with the same Poisson's ratio. The PDL was modeled as a compliant structure with a Young's modulus of 0.69 MPa and a Poisson's ratio of 0.49, as seen in Table 1. allowing simulation of realistic tissue deformation under orthodontic loading [4,21].

Tooth material, comprising enamel and dentin, was assigned a modulus of 19,600 MPa and a Poisson's ratio of 0.30. The attachment composite material was modeled with a modulus of 12,500 MPa and Poisson's ratio of 0.36. The clear aligner was simulated using multiple materials to assess biomechanical differences. Thermoformed aligners were assigned a modulus of 528 MPa and a Poisson's ratio of 0.36. These values enabled a realistic simulation of force propagation across the dentoalveolar complex, attachments, and aligner, allowing for the assessment of movement efficiency and tissue stress distributions under varying biomechanical conditions [22].

## Results

### Mesialization of upper first molar

Tooth displacement during mesialization influenced significantly by attachment shape and configuration. In single-attachment models, rectangular and trapezoid shapes produced the largest crown displacements in the mesial direction, measuring 0.0537 mm and 0.0548 mm, respectively as seen in Figs 2 and 3. However, root movement remained minimal, below 0.019 mm, indicating a high degree of tipping. Curved attachments, including ellipse and semicircle geometries, were less effective, producing crown displacements around 0.042 mm. When dual attachments were used on both buccal and lingual surfaces, crown displacement remained similar (0.054 mm), but root movement increased to 0.025 mm, resulting in a more balanced and bodily movement pattern [23].

PDL strain patterns showed the highest strain localized to the distobuccal root surface in single-attachment cases, with maximum strain values reaching 0.339 mm/mm as seen in Table 2. Dual attachments led to an increase in peak strain to 0.390 mm/mm. The strain was also more broadly distributed along the root surface. Stress in the surrounding maxillary bone peaked at 6.64 MPa near the root apex in single-attachment scenarios as seen in Table 3. The application of dual attachments increased the peak to 7.11 MPa and shifted the stress distribution apically.

### Extrusion of upper first molar

Crown displacement in extrusion simulations was highest with flat-shaped attachments. Single rectangular attachments produced a crown displacement of 0.0492 mm, while square attachments reached 0.0481 mm as seen in Figs 4 and 5. Curved attachments such as the semicircle and ellipse performed poorly, with crown movement between 0.036 and

**Table 1. Material properties and mesh size of each component of system model [4,21].**

| Material | Elastic Modulus E (MPa) | Poisson's Ratio v | Mesh Size (mm) (Tetrahedral) |
|---|---|---|---|
| Teeth | 19,600 | 0.3 | 0.6 |
| Attachment | 12,500 | 0.36 | 0.6 |
| Thermoformed Clear Aligner | 528 | 0.36 | 0.6 |
| Cortical Bone | 13,700 | 0.3 | 1.0 |
| Trabecular Bone | 1,370 | 0.3 | 1.0 |
| PDL | 0.69 | 0.45 | 0.6 |

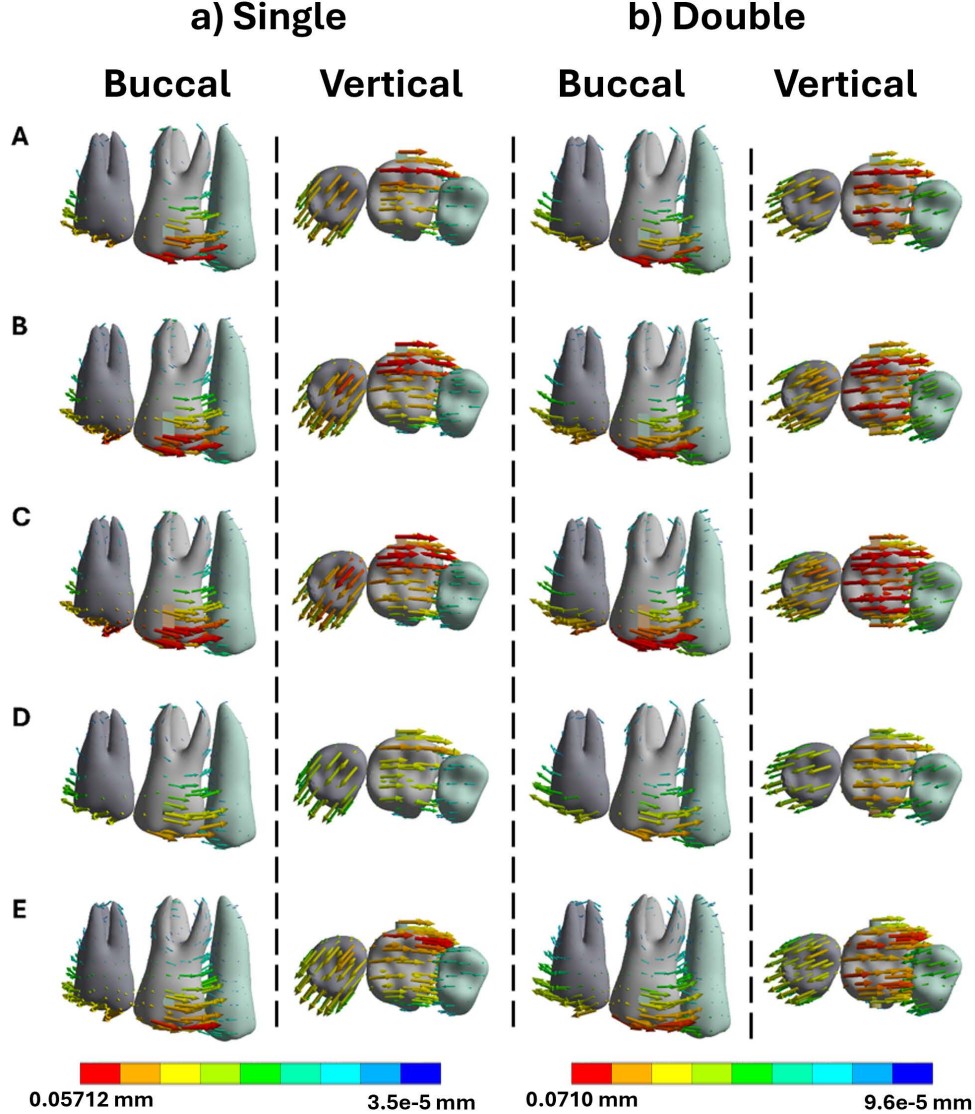

## a) Single
### Buccal    Vertical

## b) Double
### Buccal    Vertical

A

B

C

D

E

0.05712 mm          3.5e-5 mm       0.0710 mm          9.6e-5 mm

**Fig 2. Buccal and vertical views of the vector displacements in millimeters of the posterior teeth after 1st molar mesialization using single and double attachments. A)** Square, **B)** Rectangle, **C)** Trapezoid, **D)** Quad Ellipse, and **E)** Semicircle.

0.038 mm. In single-attachment setups, buccolingual asymmetry caused tilting and cusp height discrepancies of up to 0.012 mm. Dual attachments reduced this asymmetry and increased root displacement (from 0.007 mm to 0.019 mm), producing a more parallel crown–root movement [24].

PDL stress showed the highest compression in the apical third of the root with flat attachments. Strain peaked at 0.307 mm/mm for the trapezoidal design in single placement and rose to 0.353 mm/mm in dual configurations as seen in Table 2. In the bone, extrusion stresses were concentrated near the root apex. Single-attachment models recorded peak stress of 4.64 MPa, while dual attachments exhibited maximum stress of 4.97 MPa and expanded the affected region medially along the root as seen in Table 3.

## a) Single

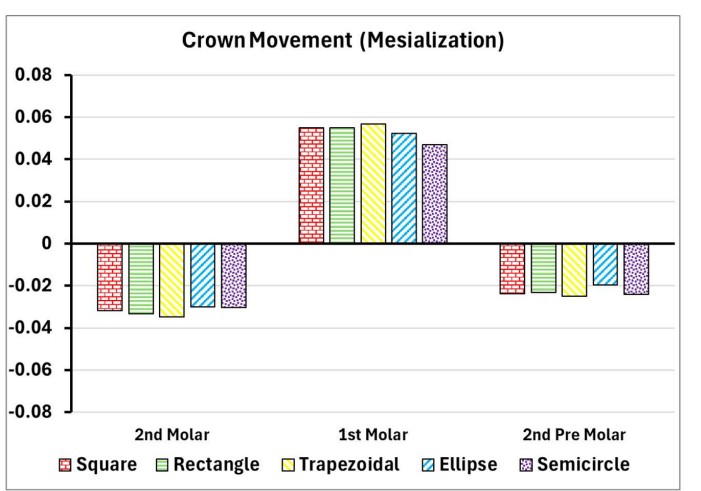

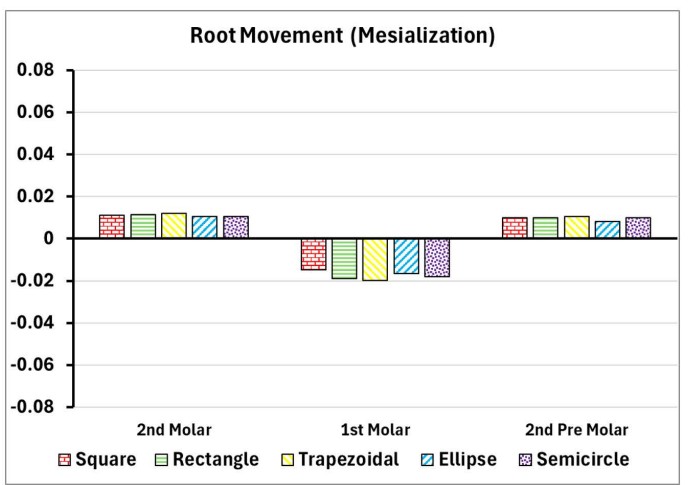

## b) Double

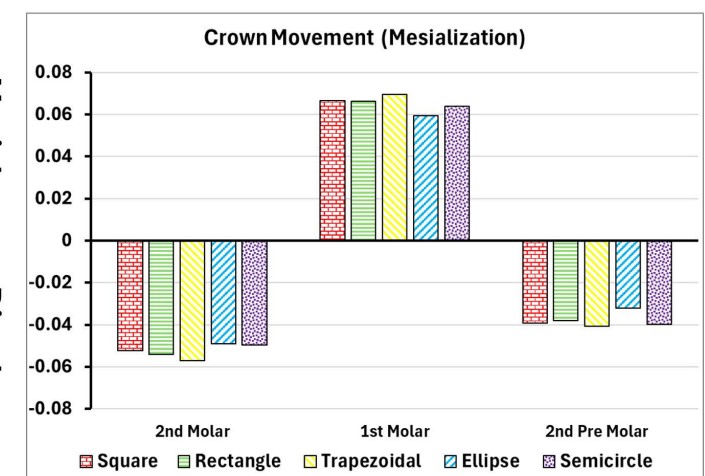

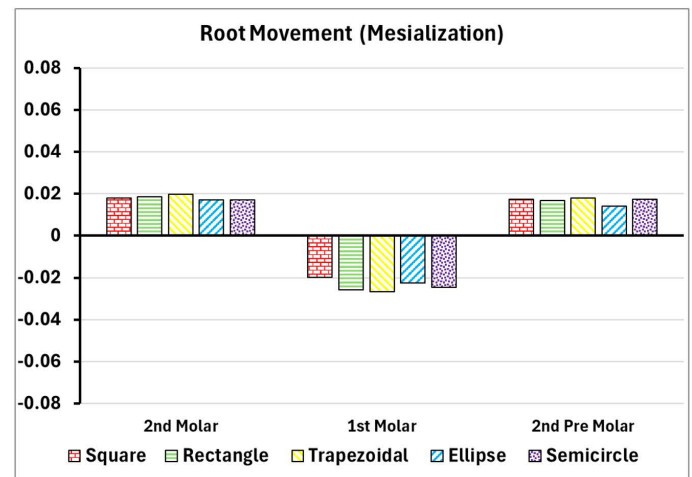

**Fig 3. Crown and root displacements in millimeters of the posterior teeth after 1st molar mesialization with single and double attachments. A)** Square, **B)** Rectangle, **C)** Trapezoid, **D)** Quad Ellipse, and **E)** Semicircle.

**Table 2. Maximum strains in the PDL (mm/mm).**

| Attachment Shape | Mesialization | | Extrusion | | Intrusion | | Rotation | |
|---|---|---|---|---|---|---|---|---|
| | Single | Double | Single | Double | Single | Double | Single | Double |
| Square | 0.328 | 0.377 | 0.303 | 0.348 | 0.052 | 0.0598 | 0.314 | 0.361 |
| Rectangle | 0.334 | 0.384 | 0.305 | 0.351 | 0.053 | 0.0605 | 0.318 | 0.365 |
| Trapezoidal | 0.339 | 0.390 | 0.307 | 0.353 | 0.054 | 0.0625 | 0.326 | 0.375 |
| Ellipse | 0.320 | 0.368 | 0.296 | 0.340 | 0.051 | 0.0588 | 0.310 | 0.356 |
| Semicircle | 0.324 | 0.374 | 0.300 | 0.345 | 0.053 | 0.0611 | 0.031 | 0.358 |

**Table 3. Maximum von Mises Stress in the bone (MPa).**

| Attachment Shape | Mesialization | | Extrusion | | Intrusion | | Rotation | |
|---|---|---|---|---|---|---|---|---|
| | Single | Double | Single | Double | Single | Double | Single | Double |
| Square | 6.43 | 6.88 | 6.90 | 4.84 | 1.73 | 1.85 | 6.47 | 6.93 |
| Rectangle | 6.42 | 6.89 | 6.88 | 4.85 | 1.72 | 1.85 | 6.49 | 6.95 |
| Trapezoidal | 6.45 | 7.11 | 7.12 | 4.97 | 1.73 | 1.86 | 6.61 | 7.07 |
| Ellipse | 6.29 | 6.72 | 6.71 | 4.81 | 1.70 | 1.81 | 6.40 | 6.85 |
| Semicircle | 6.32 | 6.75 | 6.75 | 4.85 | 1.68 | 1.83 | 6.42 | 6.89 |

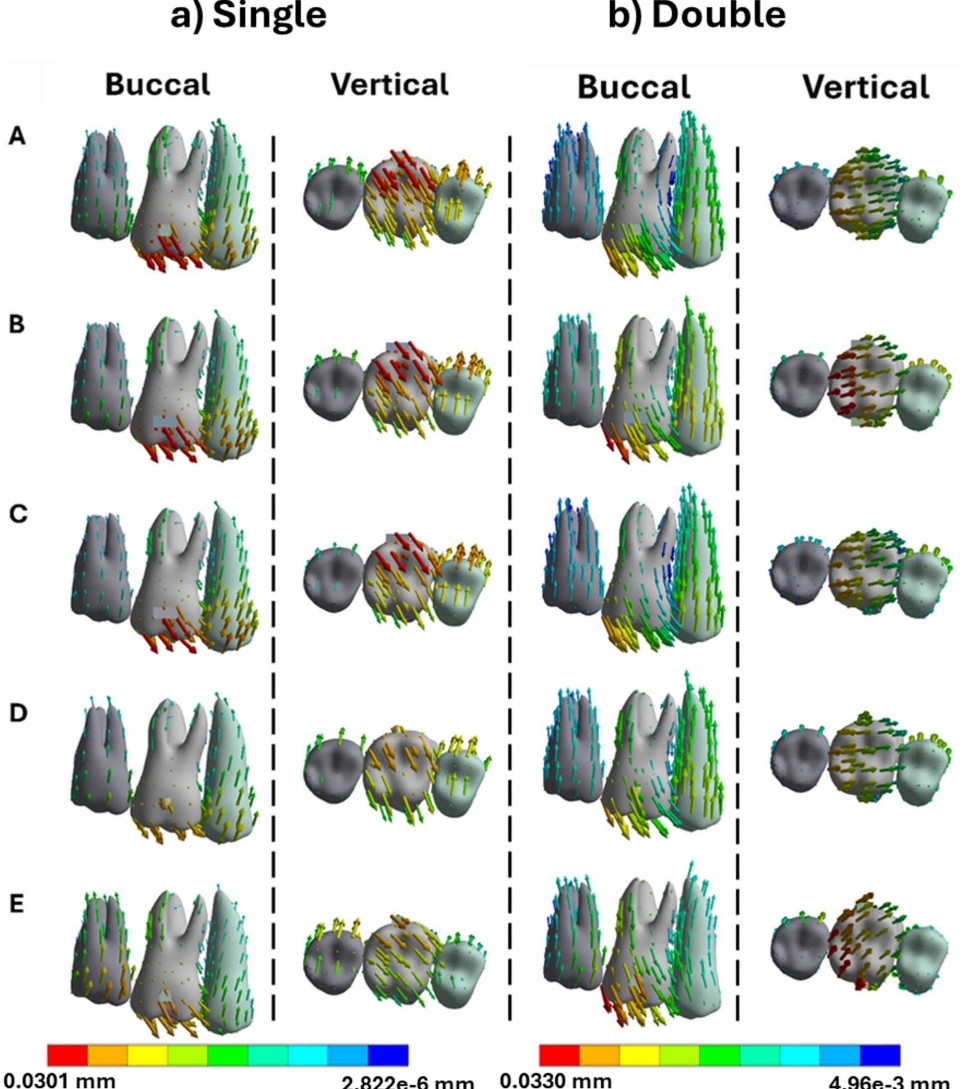

**Fig 4. Buccal and vertical views of the vector displacements in millimeters of the posterior teeth after 1st molar extrusion using single and double attachments. A)** Square, **B)** Rectangle, **C)** Trapezoid, **D)** Quad Ellipse, and **E)** Semicircle.

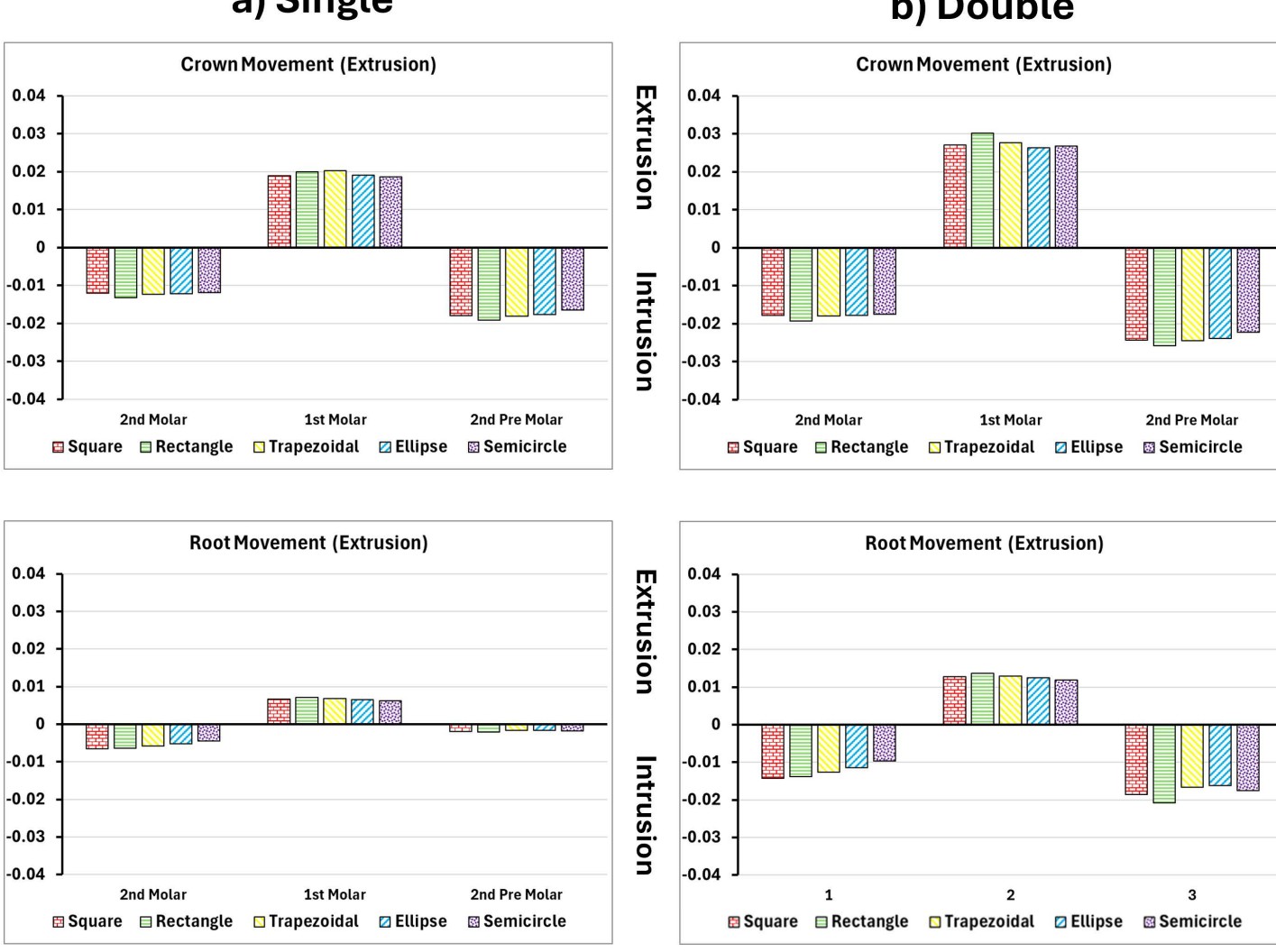

**Fig 5. Crown and root displacements in millimeters of the posterior teeth after 1ˢᵗ molar extrusion with single and double attachments. A)** Square, **B)** Rectangle, **C)** Trapezoid, **D)** Quad Ellipse, and **E)** Semicircle.

### Intrusion of upper first molar

Intrusion movements were limited, particularly with single attachments. Rectangular and trapezoid designs produced crown intrusions of 0.0391 mm and 0.0375 mm (Figs 6 and 7). Root displacement stayed below 0.011 mm, indicating tipping rather than true bodily intrusion. Dual attachments increased root movement to approximately 0.020 mm, improving vertical alignment and overall control [18].

PDL maximum strain was elevated near the apex in single-attachment simulations, with trapezoid attachments generating up to 0.054 mm/mm. Dual attachments reduced this to approximately 0.063 mm/mm as seen in Table 2. The broader strain distribution in dual configurations lowered the risk of apical overload, which is associated with root resorption. Bone stress during intrusion followed the same trend. The alveolar crest experienced a maximum of 1.73 MPa in single-attachment conditions while the dual attachments increased the stress to 1.86 MPa as seen in Table 3.

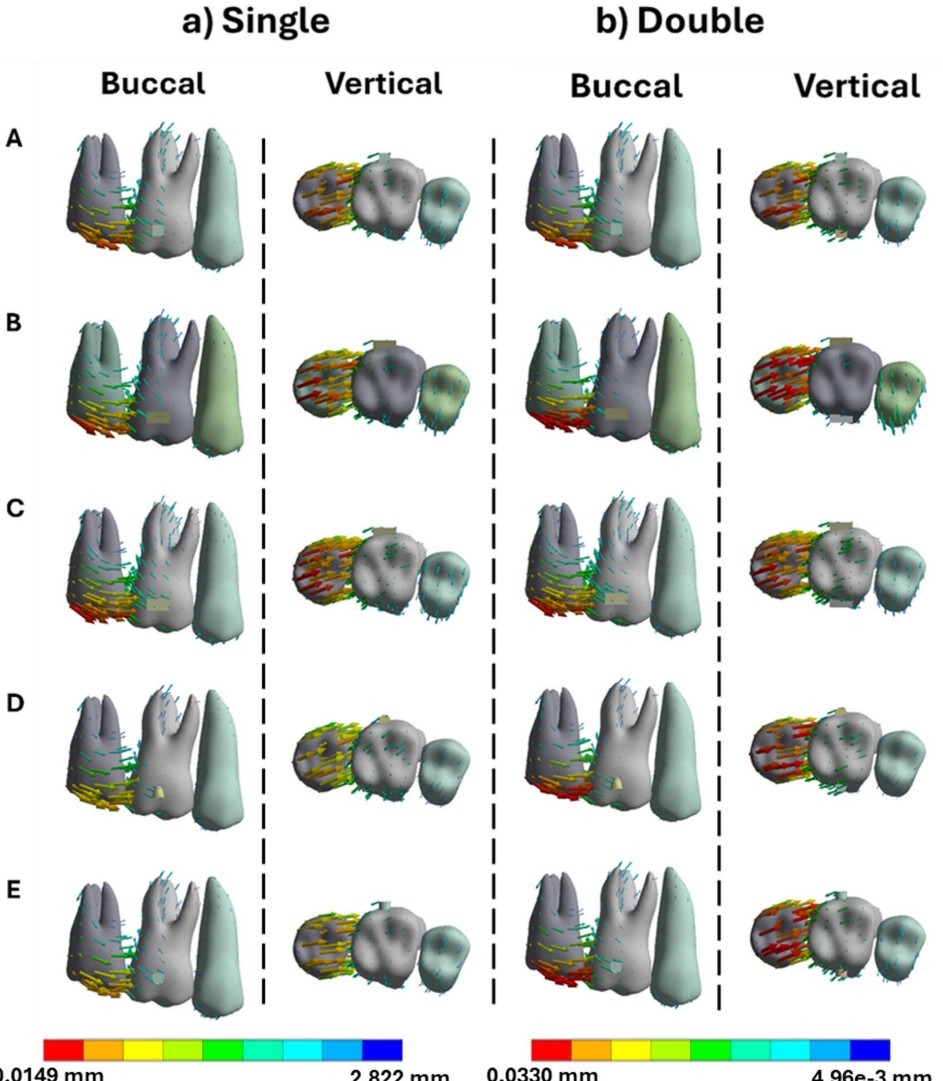

**Fig 6. Buccal and vertical views of the vector displacements in millimeters of the posterior teeth after 1ˢᵗ molar intrusion using single and double attachments. A)** Square, **B)** Rectangle, **C)** Trapezoid, **D)** Quad Ellipse, and **E)** Semicircle.

## Rotation of upper first molar

Rotational control in molar mesialization was significantly influenced by attachment shape and quantity. Single attachments led to moderate rotation, with the trapezoidal shape producing the highest rotation at 0.189°, followed by the rectangular (0.177°) and square (0.167°) attachments as seen in Figs 8 and 9, and Table 4. Curved geometries performed best in limiting unwanted rotation, with the ellipse and semicircle attachments yielding the lowest values at 0.145° and 0.151°, respectively. The introduction of double attachments consistently improved rotational control across all shapes. For instance, trapezoidal attachments showed an increase to 0.235° in rotation with dual placement, while the ellipse and semicircle designs rose slightly to 0.156° and 0.165°, respectively [9].

PDL strain peaked at 0.326 mm/mm in the buccal cervical region of single-attachment models. Double attachments peaked to 0.375 mm/mm and distributed the strain around the entire root surface, especially through the middle third as

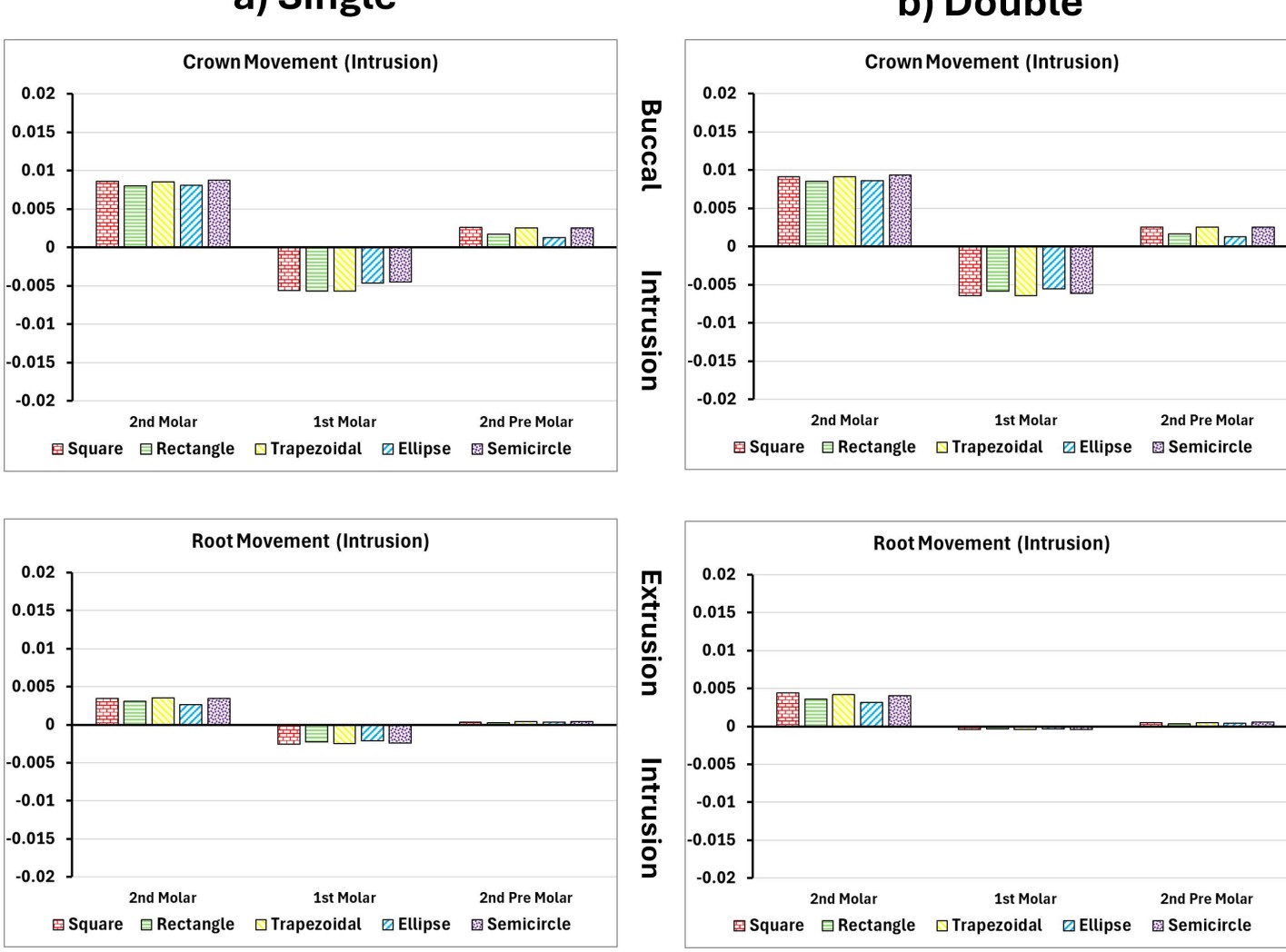

**Fig 7. Crown and root displacements in millimeters of the posterior teeth after 1ˢᵗ molar intrusion with single and double attachments. A)** Square, **B)** Rectangle, **C)** Trapezoid, **D)** Quad Ellipse, and **E)** Semicircle.

seen in Table 2. Bone stress during rotation was highly asymmetric in single-attachment models, peaking at 6.61 MPa on one side of the root. Dual attachments helped rebalance the load, increasing the peak to 7.07 MPa and eliminating sharp stress gradients near the apex as seen in Table 3.

## Discussion

This study employed a parametric finite element approach to investigate the biomechanical effects of various attachment shapes and configurations on 1st molar movement during clear aligner therapy. The analysis focused on four clinically relevant tooth movements (mesialization, extrusion, intrusion and rotation) and evaluated key biomechanical outputs including crown and root displacement, PDL strain, and maxillary bone stress. The simulations revealed that both attachment geometry and placement play pivotal roles in determining the quality and control of tooth movement [4].

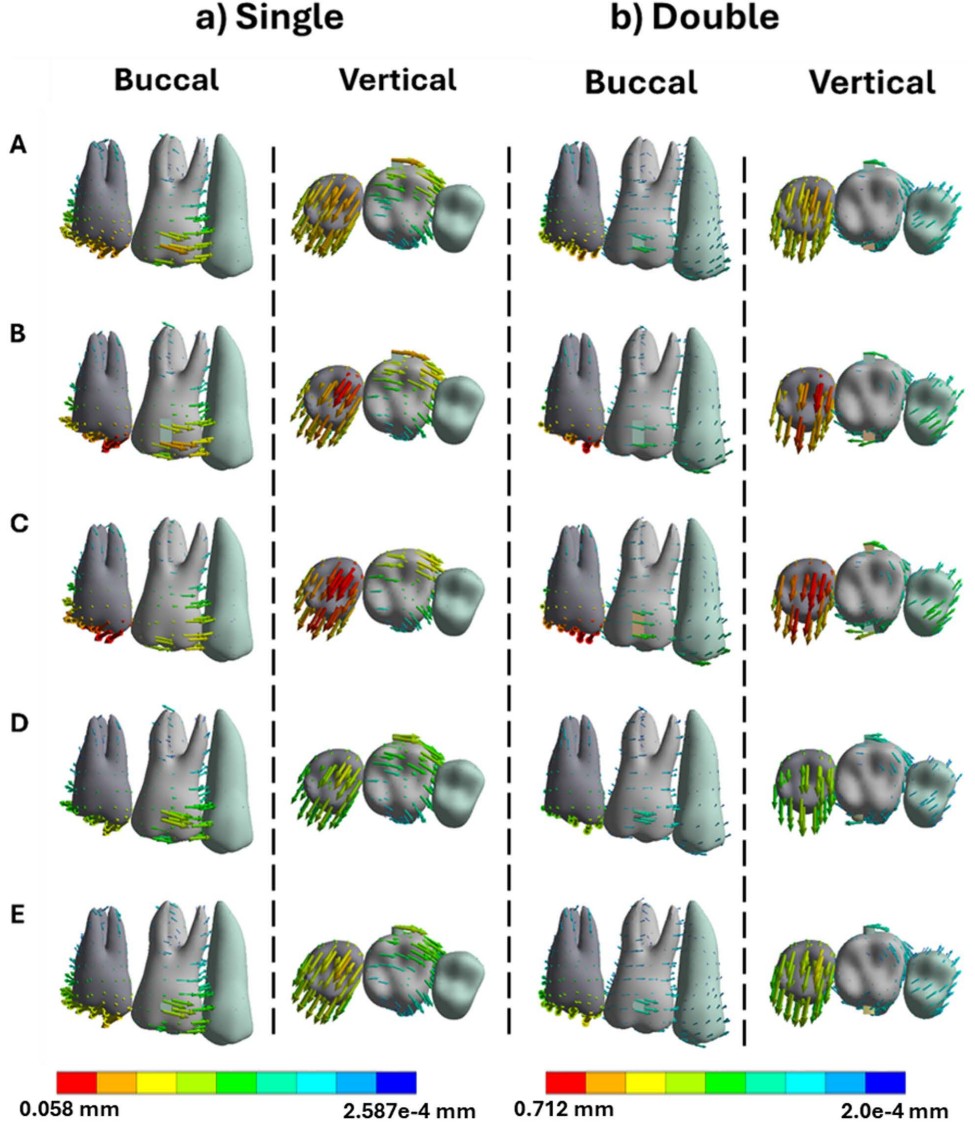

**Fig 8. Buccal and vertical views of the vector displacements in millimeters of the posterior teeth after 1st molar rotation using single and double attachments. A)** Square, **B)** Rectangle, **C)** Trapezoid, **D)** Quad Ellipse, and **E)** Semicircle.

During mesialization, trapezoidal and rectangular attachments produced the highest crown displacements (up to 0.0548 mm), with limited root translation (<0.019 mm), indicating significant tipping due to concentrated forces at the crown. In single buccal attachment setups, the unbalanced force distribution caused mesial crown movement but distal root displacement. This created a rotational moment around the tooth's center of resistance. Curved attachments such as ellipses and semicircles demonstrated lower crown displacement (0.047–0.052 mm) and exhibited reduced root control, reflecting poor performance in promoting bodily movement [25].

In contrast, dual attachments on both buccal and lingual surfaces improved root engagement and overall translation. While crown displacement remained consistent (0.054 mm), root movement increased to 0.025 mm, suggesting a more parallel, controlled displacement. PDL strain in mesialization peaked at 0.390 mm/mm in dual attachment models that

## a) Single

## b) Double

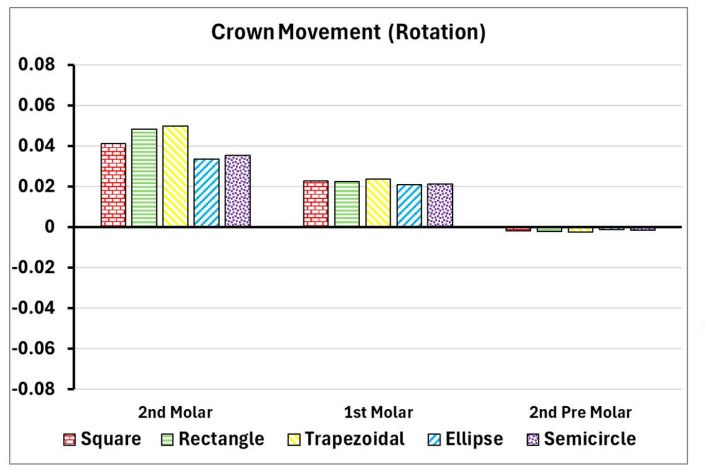

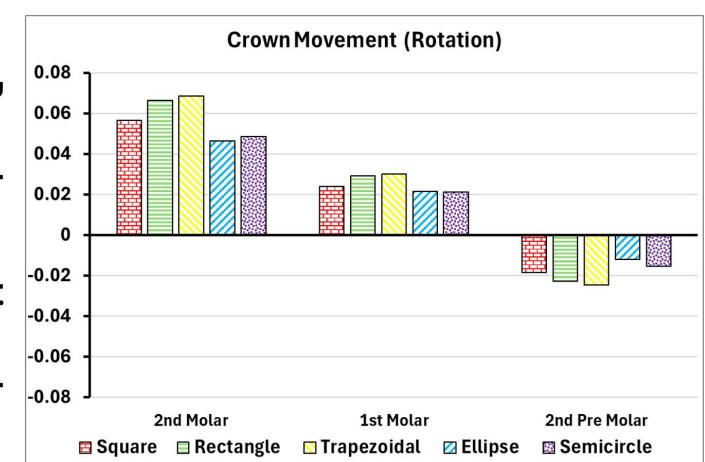

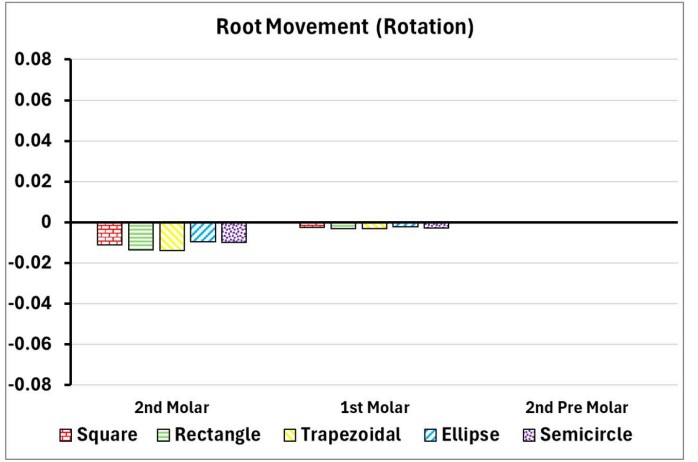

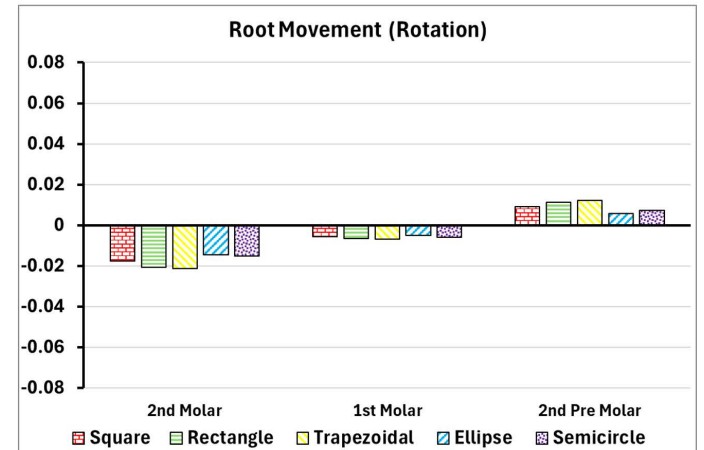

**Fig 9. Crown and root displacements in millimeters of the posterior teeth after 1st molar rotation with single and double attachments. A)** Square, **B)** Rectangle, **C)** Trapezoid, **D)** Quad Ellipse, and **E)** Semicircle.

**Table 4. Maximum rotation in the 1st molar with singular and double attachments.**

| Attachment Shape | Single | Double |
|---|---|---|
| Square | 0.167 | 0.212 |
| Rectangle | 0.177 | 0.198 |
| Trapezoidal | 0.189 | 0.235 |
| Ellipse | 0.145 | 0.156 |
| Semicircle | 0.151 | 0.165 |

were higher than the 0.339 mm/mm observed in single models but was more evenly distributed along the root surface, reducing localized overload. Bone stress also increased from 6.64 MPa to 7.11 MPa, shifting apically and indicating deeper load transmission along the root [26].

While both mesialization and distalization present similar biomechanical requirements, distalization has been investigated more extensively in both finite element and clinical aligner studies [7,27]. In contrast, mesialization remains under-explored despite its clinical importance, particularly in space closure following premolar extraction or in cases requiring anterior anchorage reinforcement. In addition, mesialization involved only a single tooth not complicated by multiple tooth movement with the periodontal ligament.

In extrusion simulations, flat-shaped attachments like rectangular and square designs demonstrated superior coronal displacement (0.049 mm), while curved attachments (semicircle, ellipse) performed poorly (0.036–0.038 mm). Single-attachment setups caused buccolingual asymmetry and tipping, with crown displacement nearly 2.8 times greater than root movement due to the attachment's position and aligner mechanics. Dual attachments helped reduce this imbalance, increasing root displacement to 0.019 mm and improving crown-root alignment. PDL strain increased from 0.307 mm/mm to 0.353 mm/mm in dual configurations, with a broader distribution pattern. Bone stress was concentrated around the root apex, increasing from 4.64 MPa in single attachments to 4.97 MPa in dual ones and spreading medially along the root surface [18].

Intrusion movements yielded the lowest displacement values across all attachment types. Crown movement was generally limited to 0.0066 mm, with root displacement remaining under 0.001 mm. Flat attachments again outperformed curved ones, but the efficiency of intrusion overall was limited by anatomical and physiological constraints. Single attachments led to minimal control and distal tipping, while dual attachments marginally improved vertical alignment. PDL strain during intrusion was the lowest of all movement types, averaging around 0.00038 mm/mm with no significant peaks. Bone stress remained modest as well, increasing from 1.73 MPa to 1.86 MPa between single and dual configurations, concentrated primarily at the alveolar crest and root socket [28].

Rotational movement showed a strong dependency on attachment shape. Trapezoidal attachments induced the greatest angular displacement (0.235°), followed by rectangular and square shapes. Curved attachments, particularly ellipse and semicircle, showed the least rotation (0.165°), demonstrating greater resistance to unwanted rotational forces. With single attachments, uneven torque produced mesial crown tipping and distal root movement. Dual attachments balanced the force system, reducing asymmetry, although they slightly increased total angular movement. PDL strain peaked at 0.375 mm/mm in dual models compared to 0.326 mm/mm in single models, with forces distributed across the middle third of the root. Bone stress rose from 6.61 MPa in single-attachment configurations to 7.07 MPa in dual attachments, with maximum values observed at the crown's distobuccal surface and furcation region of the root [27].

These findings demonstrate that attachment shape and configuration are key determinants of force transmission and biological response during aligner-based orthodontic treatments. Flat attachments (trapezoid, rectangular, square) provided the most effective displacement. However, this came at the cost of higher PDL strain and localized bone stress, especially at the crown–root interface. In contrast, curved attachments offered more diffused force delivery and lower strain concentrations but were less effective in achieving clinically desirable tooth movement, especially for complex tasks such as bodily mesialization or controlled rotation.

Dual attachments emerged as the most effective configuration for bodily control across all movement types. By distributing forces across both buccal and lingual surfaces, dual attachments improved root engagement, reduced asymmetrical tipping, and enhanced the precision of movement vectors. This was especially evident in mesialization and extrusion simulations, where dual models produced more even displacement patterns and reduced anchorage loss. However, the added force complexity resulted in slightly higher stress and strain concentrations at both the crown and root interfaces, which, if not managed properly, could increase the risk of root resorption, PDL trauma, or soft tissue irritation [7].

The biologic significance of the stresses and strains observed in our simulations can be framed within Frost's mechanostat concept. Frost described distinct strain windows governing bone adaptation: disuse (<800 µε), physiologic maintenance (800–1,500 µε), overload modeling (>1,500–3,000 µε), and pathologic overload (>4,000–5,000 µε) [29–31]. While these thresholds are expressed in bone microstrain and cannot be directly equated to the PDL strains (up to 0.39 mm/mm) or bone stresses (~7 MPa) reported here, the relative elevation observed with flat dual attachments suggests movement efficiency is achieved at the cost of approaching higher biologic loading zones. This interpretation highlights the need for clinical caution: while rectangular and trapezoidal dual attachments maximize bodily control, they may also increase the risk of localized tissue overload in susceptible patients, particularly those with reduced periodontal support or pre-existing root resorption risk.

Clinically, these results reinforce the critical role of tailored attachment design in optimizing clear aligner therapy. While larger, flatter attachments are ideal for maximizing force application, their use should be balanced against the risk of biological overload in sensitive regions such as the apical root and furcation areas. Curved attachments may be more appropriate in scenarios requiring lower force application or where tissue sensitivity is a concern. Dual attachments offer the highest control but are not commonly used in practice due to clinical limitations, including difficulty in bonding to lingual surfaces and interference from saliva, which may compromise retention [32].

Overall, attachment shape and placement should be carefully matched to the biomechanical demands of the desired tooth movement. Strategic selection can enhance movement efficiency, minimize unwanted effects like tipping and anchorage loss, and reduce the risk of long-term periodontal complications. These insights may help guide more personalized and effective treatment planning in clear aligner therapy, particularly for challenging cases involving molar movements.

Limitations of this study include its quasi-static simulation approach, which does not account for force decay, fatigue, or intraoral wear over time. Uniform aligner thickness was assumed across all models, which may not reflect clinical variability. Finally, considering the thermoformed clear aligner to be perfectly in-contact with the teeth during the treatment is another limitation of this study.

## Conclusion

In conclusion, this study evaluated the biomechanical response of different attachment shapes and placements under clear aligner-based mesialization using finite element analysis. The main findings can be summarized as follows:

- Flat-shaped attachments, such as rectangles and trapezoids, produced the highest tooth displacements but also led to greater von Mises and PDL stresses, especially near the crown and root apex.

- Curved attachments showed reduced stress concentrations and safer force distribution, though at the cost of movement efficiency, making them more appropriate in patients with biological risk factors.

- Dual attachment configurations (buccal and lingual) consistently improved root engagement, reduced tipping, and promoted bodily movement across all simulated directions.

- Single attachments resulted in higher stress asymmetry and increased risk of anchorage loss, particularly in adjacent premolars and molars.

## Author contributions

**Conceptualization:** Khaled Alsharif, Peter Ngan, Guoqiang Guan, Egon Mamboleo, Osama M. Mukdadi.

**Data curation:** Egon Mamboleo.

**Formal analysis:** Khaled Alsharif, Peter Ngan, Egon Mamboleo, Osama M. Mukdadi.

**Funding acquisition:** Osama M. Mukdadi.

**Investigation:** Khaled Alsharif, Peter Ngan, Guoqiang Guan, Egon Mamboleo, Ali Merdji, Osama M. Mukdadi.

**Methodology:** Egon Mamboleo, Abdelhak Ouldyerou, Osama M. Mukdadi.

**Project administration:** Osama M. Mukdadi.

**Resources:** Osama M. Mukdadi.

**Software:** Egon Mamboleo, Osama M. Mukdadi.

**Supervision:** Osama M. Mukdadi.

**Validation:** Egon Mamboleo, Osama M. Mukdadi.

**Visualization:** Abdelhak Ouldyerou, Osama M. Mukdadi.

**Writing – original draft:** Egon Mamboleo, Osama M. Mukdadi.

**Writing – review & editing:** Khaled Alsharif, Peter Ngan, Guoqiang Guan, Egon Mamboleo, Ali Merdji, Osama M. Mukdadi.

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
