## [Decision Letter · Decision Letter 0]

1 Aug 2025

Dear Dr. Mukdadi,

We look forward to receiving your revised manuscript.

Kind regards,

Mohmed Isaqali Karobari, BDS, MScD.Endo, Ph.D. Endo, FDS, FPFA, FICD, MFDS

Academic Editor

PLOS ONE

Journal Requirements:

3. We note that your Data Availability Statement is currently as follows: All relevant data are within the manuscript and in Supporting Information files.

Additional Editor Comments:

Dear Authors,

Kindly read all the comments given by the reviewers carefully and address them; make the changes in the revised manuscript accordingly.

Best regards and keep well

Reviewers' comments:

Reviewer's Responses to Questions

**Comments to the Author**

1. Is the manuscript technically sound, and do the data support the conclusions?

Reviewer #1: Yes

Reviewer #2: Yes

2. Has the statistical analysis been performed appropriately and rigorously?

Reviewer #1: Yes

Reviewer #2: N/A

3. Have the authors made all data underlying the findings in their manuscript fully available?

Reviewer #1: Yes

Reviewer #2: Yes

4. Is the manuscript presented in an intelligible fashion and written in standard English?

Reviewer #1: Yes

Reviewer #2: Yes

Reviewer #1: Title: Effects of attachment designs on clear aligner tooth movement: a finite element analysis

This manuscript, entitled “Effects of Attachment Designs on Clear Aligner Tooth Movement: A Finite Element Analysis”, aims to evaluate the biomechanical impact of various attachment shapes and configurations on maxillary molar movement using finite element modeling. By simulating mesialization, intrusion, extrusion, and rotation, the study provides valuable insights into how attachment geometry and placement influence displacement patterns, PDL strain, and bone stress.

The findings suggest that flat attachments (e.g., rectangular and trapezoidal) enhance tooth movement efficiency but increase localized stress, while curved attachments distribute forces more evenly with less effective movement. Dual attachment configurations showed improved root control and bodily displacement. Overall, the manuscript is well-organized and methodologically sound. However, several areas would benefit from clarification or expansion.

Clarity of Scope

The introduction provides a strong biomechanical rationale for studying different attachment shapes and configurations. However, the clinical relevance of the selected movements could be better contextualized. Notably, molar distalization, one of the most commonly targeted movements in aligner therapy, is not included. The authors should clarify whether this was a modeling limitation or an intentional scope decision. Alternatively, the title and abstract could be revised to reflect the primary focus on mesialization and its biomechanical implications.

Methods

The finite element modeling approach is detailed and robust. However, the simulation of a single molar limits extrapolation to full-arch mechanics. The authors are encouraged to discuss the potential impact of this simplification on the generalizability of results to multi-tooth clinical settings.

Results and Discussion

• The results are clearly presented, but a more explicit clinical interpretation of PDL strain and bone stress thresholds would add practical value.

• The discussion of buccal-lingual dual attachments would benefit from addressing the clinical challenges of lingual bonding and aligner retention.

• The manuscript would be strengthened by referencing recent FEA studies (e.g., Gao et al. 2023; Hong et al. 2021), which offer relevant comparative insight.

• The omission of distalization should be explicitly justified, considering its clinical relevance in aligner-based therapies.

• The manuscript is well written. However, some long and technical sentences, particularly in the Results and Discussion sections, could benefit from slight editing to improve clarity and readability.

Modeling Assumptions and Opportunities for Enhancement

• The authors mention that a mesh convergence study was performed, but do not include numerical data or describe the specific procedure. A brief summary or convergence plot (e.g., stress vs. mesh size) would improve transparency and validate the numerical stability of the model.

• The simulation applies simplified static loading. Could the authors explain why dynamic approaches (e.g., birth and death method) were not considered? These can provide more realistic insight into aligner deformation and force delivery.

• Unlike other recent FEA studies, this manuscript does not include a baseline simulation without attachments. Including a no-attachment model would allow for a clearer evaluation of the true benefit of each attachment design. Was this omission due to a modeling constraint or an intentional choice?

• The aligner thickness is fixed at 0.8 mm but not justified. Given the known impact of thickness on stiffness and force transmission, the rationale should be provided, and ideally, sensitivity to different thicknesses tested.

Limitations Section

While the discussion touches on some modeling constraints, the manuscript would benefit from a dedicated limitations paragraph, highlighting key assumptions (e.g., isotropic material properties, static load conditions, single-subject modeling, exclusion of time-dependent tissue response).

Clarify Clinical Relevance

The simulations are methodologically sound, but the clinical applicability of each scenario could be better emphasized. For instance, how often are dual attachments feasible in real clinical practice? Are there clinical data supporting the use of the attachment shapes tested here? Framing the biomechanical findings in a clinical context would increase the practical impact of the study.

Moreover, the authors could better highlight how this study adds novel value beyond previous FEA studies on attachments. What specific insight does this work contribute to clinical treatment planning or future appliance design?

Final Notes to the Authors

This manuscript presents a detailed finite element analysis of different attachment geometries used in clear aligner therapy, with well-structured methodology and valuable biomechanical findings. However, several important aspects should be addressed to improve clarity, completeness, and clinical relevance. Clarifying modeling decisions, referencing recent comparative work, justifying design choices (e.g., thickness), and strengthening the clinical framing will significantly enhance the contribution of this study to the field.

Reviewer #2: The objective of this study is to investigate the biomechanical effects of different attachment shapes and configurations on the movement, stress, and strain distribution of the maxillary first molar during clear aligner therapy by using FEM. This study holds scientific merit, and the authors deserve recognition for their diligent work. However, several key issues need to be addressed before the manuscript can be considered for publication.

1. Regarding the finite element simulation, the manuscript needs a clear statement defining the accuracy of the FE model. The reviewer thought that the FE model used in this study needs to be validated before use. Alternatively, please consider adding one paragraph to the “Discussion” section that describes the reliability of the FE model used in this study.

2. In clinical contexts, the human PDL is not a linear elastic material. The authors have modeled the PDL as linear elastic with Elastic Modulus of 0.69 MPa, which does not fully reflect real physiological conditions. Studies exist that model PDL as a bilinear elastic material, which more accurately represents the properties of human PDL. The authors are encouraged to expand on the simplifications made regarding PDL material properties and to compare their approach with other studies using bilinear elastic models, such as:

Effects of Attachment Design and Aligner Material on Mandibular Canine Distal Bodily Movement in Aligner Treatment.

https://link.springer.com/article/10.1007/s40846-024-00904-5

Determination of the elasticity parameters of the human periodontal ligament and the location of the center of resistance of single-rooted teeth a study of autopsy specimens and their conversion into finite element models https://pubmed.ncbi.nlm.nih.gov/12297965/

The reviewer recommends that a paragraph be added in the Discussion to address the impact of PDL material assumptions on the results, or at the very least, to acknowledge this point as a limitation of the present study.

3. In the sentence: “The periodontal ligament (PDL) was reconstructed by isolating the tooth roots and extruding them outward by 0.2 mm, consistent with anatomical thicknesses reported in literature [11],”

it is suggested to simply use “PDL” instead of repeating the full term “The periodontal ligament (PDL),” since the abbreviation has already been defined earlier in the manuscript. This will improve conciseness and readability.

**Do you want your identity to be public for this peer review?** For information about this choice, including consent withdrawal, please see our Privacy Policy

Reviewer #1: No

Reviewer #2: No

---

## [Author Response · Author response to Decision Letter 1]

25 Nov 2025

Reviewer #1:

Title: Effects of attachment designs on clear aligner tooth movement: a finite element analysis

This manuscript, entitled “Effects of Attachment Designs on Clear Aligner Tooth Movement: A Finite Element Analysis”, aims to evaluate the biomechanical impact of various attachment shapes and configurations on maxillary molar movement using finite element modeling. By simulating mesialization, intrusion, extrusion, and rotation, the study provides valuable insights into how attachment geometry and placement influence displacement patterns, PDL strain, and bone stress. The findings suggest that flat attachments (e.g., rectangular and trapezoidal) enhance tooth movement efficiency but increase localized stress, while curved attachments distribute forces more evenly with less effective movement. Dual attachment configurations showed improved root control and bodily displacement. Overall, the manuscript is well-organized and methodologically sound. However, several areas would benefit from clarification or expansion.

Response: Thank you for your constructive comments and the manuscript has been modified to meet your comments accordingly.

Clarity of Scope

The introduction provides a strong biomechanical rationale for studying different attachment shapes and configurations. However, the clinical relevance of the selected movements could be better contextualized. Notably, molar distalization, one of the most commonly targeted movements in aligner therapy, is not included. The authors should clarify whether this was a modeling limitation or an intentional scope decision. Alternatively, the title and abstract could be revised to reflect the primary focus on mesialization and its biomechanical implications.

Response: Thank you for the valuable comment. We have chosen single tooth movement so as not to include multiple teeth such as the second molar in molar distalization . In this study, four different tooth movement was included (mesialization, intrusion, extrusion, and rotation of the maxillary first molar). Ongoing studies by our team are focused on tooth space-closure where both mesialization and distalization will be presented in future studies. The body of the manuscript has been revised to meet your comment. The authors agreed that the title is clear and suitable for the work presented in this manuscript.

Methods

The finite element modeling approach is detailed and robust. However, the simulation of a single molar limits extrapolation to full-arch mechanics. The authors are encouraged to discuss the potential impact of this simplification on the generalizability of results to multi-tooth clinical settings.

Response: A single tooth model was used as a computational benchmark model to assess the efficacy of attachment shape and design on tooth movement. This simplified single tooth model provided a means to eliminate complex factors that would interfere with the effectiveness of each attachment design and placement. Future studies with multiple tooth movement will be used to generalize for full-arch mechanics. Thanks for the suggestion.

Results and Discussion

• The results are clearly presented, but a more explicit clinical interpretation of PDL strain and bone stress thresholds would add practical value.

Response: The biologic significance of the stresses and strains observed in our simulations can be framed within Frost’s mechanostat concept. Frost described distinct strain windows governing bone adaptation: disuse (<800 µε), physiologic maintenance (800–1,500 µε), overload modeling (>1,500–3,000 µε), and pathologic overload (>4,000–5,000 µε) (Frost, 1994; Frost, 2003; Frost, 2004). While these thresholds are expressed in bone microstrain and cannot be directly equated to the PDL strains (up to 0.39 mm/mm) or bone stresses (~7 MPa) reported here, the relative elevation observed with flat dual attachments suggests movement efficiency is achieved at the cost of approaching higher biologic loading zones. This interpretation highlights the need for clinical caution: while rectangular and trapezoidal dual attachments maximize bodily control, they may also increase the risk of localized tissue overload in susceptible patients, particularly those with reduced periodontal support or pre-existing root resorption risk.

References:

Frost HM. Wolff’s Law and bone’s structural adaptations to mechanical usage: an overview for clinicians. Angle Orthod. 1994;64(3):175–188.

Frost HM. A 2003 update of bone physiology and Wolff’s Law for clinicians. Angle Orthod. 2004;74(1):3–15.

Frost HM. Bone’s mechanostat: A 2003 update. Anat Rec Part A. 2003;275(2):1081–1101.

• The discussion of buccal-lingual dual attachments would benefit from addressing the clinical challenges of lingual bonding and aligner retention.

Response: While our finite element results show that dual buccal–lingual attachments improve bodily control, clinical application is constrained by practical considerations. Lingual bonding has been reported as technically challenging due to limited access, saliva contamination, and reduced enamel surface area. Furthermore, lingual attachments may compromise patient comfort, interfere with speech, and increase aligner undercuts, potentially reducing long-term retention. These limitations help explain why dual attachments, though biomechanically effective, remain uncommon in routine clinical practice

• The manuscript would be strengthened by referencing recent FEA studies (e.g., Gao et al. 2023; Hong et al. 2021), which offer relevant comparative insight.

Response: Thank you for your suggestion, these references are now included in the revised version.

• The omission of distalization should be explicitly justified, considering its clinical relevance in aligner-based therapies.

Response: While both mesialization and distalization present similar biomechanical requirements, distalization has been investigated more extensively in both finite element and clinical aligner studies (Zhu et al., 2023; Miao et al., 2025). In contrast, mesialization remains underexplored despite its clinical importance, particularly in space closure following premolar extraction or in cases requiring anterior anchorage reinforcement. In addition, mesialization involved only a single tooth not complicated by multiple tooth movement with the periodontal ligament.

• The manuscript is well written. However, some long and technical sentences, particularly in the Results and Discussion sections, could benefit from slight editing to improve clarity and readability.

Response: We appreciate the reviewer’s feedback regarding readability. We have revised the manuscript to eliminate multiple long and technical sentences in the Results and Discussion sections for clarity. Thanks for the comments.

Modeling Assumptions and Opportunities for Enhancement

• The authors mention that a mesh convergence study was performed, but do not include numerical data or describe the specific procedure. A brief summary or convergence plot (e.g., stress vs. mesh size) would improve transparency and validate the numerical stability of the model.

Response: The accuracy of the FE model was determined by evaluating the error in calculating maximum stress and maximum nodal displacement within the model. A 5% error threshold was set to evaluate FE simulation accuracy and convergence. Convergence was confirmed when subsequent refinements led to a convergence error of 4.87% in estimating maximum von Mises stresses, and a convergence error of 0.84% in calculating maximum nodal displacements [16].

The text was modified to address this comment and make it more clear to the readers, see page 5.

• The simulation applies simplified static loading. Could the authors explain why dynamic approaches (e.g., birth and death method) were not considered? These can provide more realistic insight into aligner deformation and force delivery.

Response: This simulation is using an iterative Newton-Raphson method for an initial static tooth displacement. It is already a time-consuming process, in particular with all frictional surfaces and contact elements used in this model. We agree with the reviewer that it would be better to use dynamic simulations using birth-death element method to monitor the treatment progression, however, this task is still under investigation by our research group.

• Unlike other recent FEA studies, this manuscript does not include a baseline simulation without attachments. Including a no-attachment model would allow for a clearer evaluation of the true benefit of each attachment design. Was this omission due to a modeling constraint or an intentional choice?

Response: The scope of this study is to compare various attachment designs and shapes. A model with no attachment was not considered in this study.

• The aligner thickness is fixed at 0.8 mm but not justified. Given the known impact of thickness on stiffness and force transmission, the rationale should be provided, and ideally, sensitivity to different thicknesses tested.

Response: Indeed, the aligner thickness and material properties will have an effect, yet the study is focused on the attachments rather than the aligner geometry and material properties. This aspect will be included in future studies and thank you for the suggestion.

Limitations Section

While the discussion touches on some modeling constraints, the manuscript would benefit from a dedicated limitations paragraph, highlighting key assumptions (e.g., isotropic material properties, static load conditions, single-subject modeling, exclusion of time-dependent tissue response).

Response: A paragraph on limitations has been added to the end of the discussion section.

“Limitations of this study include its quasi-static simulation approach, which does not account for force decay, fatigue, or intraoral wear over time. Uniform aligner thickness was assumed across all models, which may not reflect clinical variability. Finally, considering the thermoformed clear aligner to be perfectly in-contact with the teeth during the treatment is another limitation of this study. ”

Clarify Clinical Relevance

The simulations are methodologically sound, but the clinical applicability of each scenario could be better emphasized. For instance, how often are dual attachments feasible in real clinical practice? Are there clinical data supporting the use of the attachment shapes tested here? Framing the biomechanical findings in a clinical context would increase the practical impact of the study.

Moreover, the authors could better highlight how this study adds novel value beyond previous FEA studies on attachments. What specific insight does this work contribute to clinical treatment planning or future appliance design?

Response: Our current study found that placing attachments on both buccal and lingual surfaces resulted in better control of crown and root movements. However, clinically, this presented difficulties to clinicians who tried to bond attachment on the lingual surface due to saliva contamination. Different shapes of attachments have been used by different companies for better tooth movement (Gao et al, 2023; Zhu et al 2023 Miao et al, 2025). However, there is no consensus on whether which type of attachment design will provide better stress and strain to the periodontal ligament and localized bone tissues. The current study is an attempt to provide new data to this clinical question.

Final Notes to the Authors

This manuscript presents a detailed finite element analysis of different attachment geometries used in clear aligner therapy, with well-structured methodology and valuable biomechanical findings. However, several important aspects should be addressed to improve clarity, completeness, and clinical relevance. Clarifying modeling decisions, referencing recent comparative work, justifying design choices (e.g., thickness), and strengthening the clinical framing will significantly enhance the contribution of this study to the field.

Response: Thank you for your constructive feedback and comments. The manuscript was revised to address your comments and recent comparative work was referenced as suggested.

Reviewer #2:

The objective of this study is to investigate the biomechanical effects of different attachment shapes and configurations on the movement, stress, and strain distribution of the maxillary first molar during clear aligner therapy by using FEM. This study holds scientific merit, and the authors deserve recognition for their diligent work. However, several key issues need to be addressed before the manuscript can be considered for publication.

Response: Thank you for your time and effort in reviewing our manuscript, and for your constructive comments that definitely improved the overall quality of the manuscript.

1. Regarding the finite element simulation, the manuscript needs a clear statement defining the accuracy of the FE model. The reviewer thought that the FE model used in this study needs to be validated before use. Alternatively, please consider adding one paragraph to the “Discussion” section that describes the reliability of the FE model used in this study.

Response: The accuracy of the FE model was determined by evaluating the error in calculating maximum stress and maximum nodal displacement within the model. A 5% error threshold was set to evaluate FE simulation accuracy and convergence. Convergence was confirmed when subsequent refinements led to a convergence error of 4.87% in estimating maximum von Mises stresses, and a convergence error of 0.84% in calculating maximum nodal displacements [14].

The text was modified to address this comment and make it more clear to the readers, see page 5.

2. In clinical contexts, the human PDL is not a linear elastic material. The authors have modeled the PDL as linear elastic with Elastic Modulus of 0.69 MPa, which does not fully reflect real physiological conditions. Studies exist that model PDL as a bilinear elastic material, which more accurately represents the properties of human PDL. The authors are encouraged to expand on the simplifications made regarding PDL material properties and to compare their approach with other studies using bilinear elastic models, such as:

- Effects of Attachment Design and Aligner Material on Mandibular Canine Distal Bodily Movement in Aligner Treatment.

https://link.springer.com/article/10.1007/s40846-024-00904-5

- Determination of the elasticity parameters of the human periodontal ligament and the location of the center of resistance of single-rooted teeth a study of autopsy specimens and their conversion into finite element models https://pubmed.ncbi.nlm.nih.gov/12297965/

The reviewer recommends that a paragraph be added in the Discussion to address the impact of PDL material assumptions on the results, or at the very least, to acknowledge this point as a limitation of the present study.

Response: In this study, a linear elastic material model was adopted for the PDL model to assess the effect of the attachment design on the tooth movement. Earlier published work used bilinear or viscoelastic constitutive models. In general, the elasticity of the PDL may alter the tooth mobility and change the center resistance of the tooth. This study was focused on the attachment types and design more than the PDL model itself, and served as a comparative. Poppe et al (2002) reported that reduced periodontal tissues influenced the location of the center of resistance, which is further apical than of the healthy periodontal tissues. Therefore, we believe that adopting bilinear or viscoelastic constitutive for the PDL will not change the outcome of this comparative study. We modified the discussion to include this limitation.

3. In the sentence: “The periodontal ligament (PDL) was reconstructed by isolating the tooth roots and extruding them outward by 0.2 mm, consistent with anatomical thicknesses reported in literature [11],” it is suggested to simply use “PDL” instead of repeating the full term “The periodontal ligament (PDL),” since the abbreviation has already been defined earlier in the manuscript. This will improve conciseness and readability.

Response: We introduced the term periodontal ligament (PDL) at first appearan

---

## [Decision Letter · Decision Letter 1]

4 Mar 2026

EFFECTS OF ATTACHMENT DESIGNS ON CLEAR ALIGNER TOOTH MOVEMENT: A FINITE ELEMENT ANALYSIS

PONE-D-25-30810R1

Dear Dr. Mukdadi,

We’re pleased to inform you that your manuscript has been judged scientifically suitable for publication and will be formally accepted for publication once it meets all outstanding technical requirements.

Kind regards,

James J Cray Jr., Ph.D.

Academic Editor

PLOS One

Additional Editor Comments (optional):

Reviewers' comments:

Reviewer's Responses to Questions

**Comments to the Author**

Reviewer #1: All comments have been addressed

Reviewer #2: All comments have been addressed

2. Is the manuscript technically sound, and do the data support the conclusions?

Reviewer #1: Yes

Reviewer #2: Yes

3. Has the statistical analysis been performed appropriately and rigorously?

Reviewer #1: Yes

Reviewer #2: N/A

4. Have the authors made all data underlying the findings in their manuscript fully available?

Reviewer #1: No

Reviewer #2: Yes

5. Is the manuscript presented in an intelligible fashion and written in standard English?

Reviewer #1: Yes

Reviewer #2: Yes

Reviewer #1: I would like to thank the authors for their careful and thorough responses to all the comments raised during the review process. The revisions made have strengthened the manuscript, improving its clarity and overall scientific quality. I believe the authors have successfully addressed all concerns, and the revised version represents a clear improvement over the original submission.

Reviewer #2: (No Response)

**Do you want your identity to be public for this peer review?** For information about this choice, including consent withdrawal, please see our Privacy Policy

Reviewer #1: No

Reviewer #2: No

---

## [Editor Report · Acceptance letter]

PONE-D-25-30810R1

PLOS One

Dear Dr. Mukdadi,

I'm pleased to inform you that your manuscript has been deemed suitable for publication in PLOS One. Congratulations! Your manuscript is now being handed over to our production team.

Kind regards,

on behalf of

Dr. James J Cray Jr.

Academic Editor

PLOS One